# ESSνSB+ Target Station Concept [†]

**Tamer Tolba** [1,*] and **Eric Baussan** [2]

1   Institut for Experimental Physics, Department of Physics, Faculty of Mathematics, Informatic and Natural Sciences, Hamburg University, Luruper Chaussee 149, 22761 Hamburg, Germany
2   Université de Strasbourg, CNRS, IPHC UMR 7178, 67037 Strasbourg, France
*   Correspondence: tamer.tolba@uni-hamburg.de; Tel.: +49-40-8998-4872
†   Presented at the 23rd International Workshop on Neutrinos from Accelerators, Salt Lake City, UT, USA, 30–31 July 2022.

**Abstract:** In the search for the CP violation (CPV) in the leptonic sector, crucial information was obtained a decade ago from reactor and accelerator experiments. The discovery and measurement of the third neutrino mixing angle, $\theta_{13}$, with a value $\sim 9°$, allow for the possibility to discover the leptonic Dirac CP-violating angle, $\delta_{CP}$, with long baseline neutrino Super Beams. ESSνSB is a long-baseline neutrino project that will be able to measure the CPV in the leptonic sector at the second oscillation maximum, where the sensitivity of the experiment is higher compared to that at the first oscillation maximum. The extension project, ESSνSB+, aims to address a very challenging task on measuring the neutrino–nucleon cross-section, which is the dominant term of the systematic uncertainty, in the energy range 0.2–0.6 GeV, using a Low-Energy nuSTORM (LEnuSTORM) and an ENUBET-like Low-Energy Monitored Neutrino Beam (LEMNB) facilities. The target station plays the main role in generating a well defined and focused pion, and hence muon, beam.

**Keywords:** ESSνSB+; target station; pion beam; LEnuSTORM; LEMNB

## 1. Introduction

The present generation of ν-oscillation experiments [1,2] promises to determine the values of the three mixing angles and the two mass-splitting parameters. However, neither of these experiments can reach the confidence level for CPV discovery. The relatively high value of $\theta_{13}$ significantly modified the optimal strategy for leptonic CPV discovery and precision measurement, provided that the far detector is placed at the second, rather than the first, oscillation maximum. The future long-baseline detectors such as the Deep Underground Neutrino Experiment (DUNE) [3] in the USA, the Tokai to Hyper-Kamiokande (T2HK) [4] in Japan and the European Spallation Source neutrino Super Beam (ESSνSB) [5] in Sweden, Europe, plan to use an intense "super" neutrino beam working at mega Watt scale to reach this goal.

The recently published ESSνSB CDR [6] has demonstrated that the initially foreseen physics performance of the detector has surpassed all earlier expectations. ESSνSB will be measuring at the second oscillation maximum, where the sensitivity of CPV will be close to three times larger as compared to DUNE and T2HK experiments, which aim for measuring at the first oscillation maximum. However, the measurement of the neutrino–nucleus cross-sections (ν-A) is fundamental to reduce the systematic error in the determination of the $\delta_{CP}$ from ν-oscillations, especially since data on ν-A cross-sections in the neutrino energy range of ESSνSB, 0.2–0.6 GeV are currently very scarce [7]. An extension application of the ESSνSB project, the ESSνSB+, was submitted in March 2022 to the HORIZON-EU European Commission—and has recently been accepted (July 2022)—to be funded for the next four years. The ESSνSB+ project aims to address this very challenging task on measuring the ν-A cross-sections using a Low-Energy nuSTORM [8] (LEnuSTORM) and an ENUBET-like [9] Low-Energy Monitored Neutrino Beam (LEMNB) facilities. However, several physics and

technological challenges must be precisely studied and simulated before addressing the final design of this experiment. Among these challenges, the special target station plays the main role in producing a well-defined and focused pion beam, which will be then decayed to a muon beam in the straight section of the LEnuSTORM race track ring. Figure 1 shows the layout of the ESS site with the proposed ESS$\nu$SB and ESS$\nu$SB+ modifications.

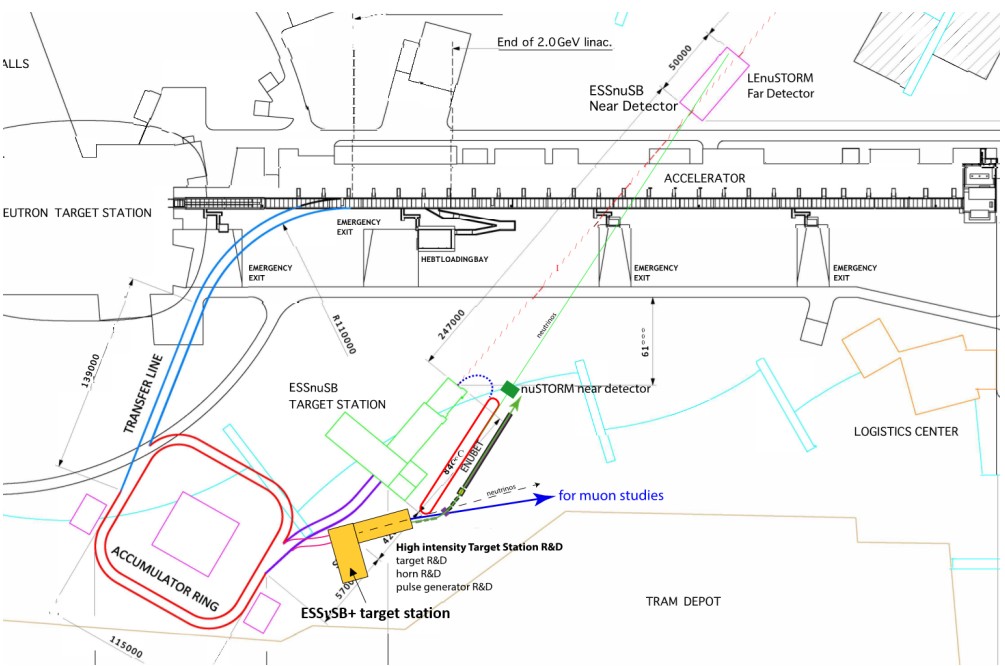

**Figure 1.** The ESS layout with the proposed modifications for the ESS$\nu$SB and ESS$\nu$SB+ experiments.

## 2. The ESS$\nu$SB Concept

As discussed here [10], the ESS linac will provide an average proton beam power of 5 MW to the spallation neutron target divided into 14 macro-pulses per second, each with a duration of 2.86 ms. Interleaved with these pulses will be another 5 MW of beam power delivered to the accumulator ring and later to the neutrino target of the ESS$\nu$SB target station facility, also at 14 Hz. Those additional proton pulses will be brought from the end of the linac by a ring-to-switchyard (R2S) beam transfer line to a beam switchyard (BSY), which will share and focus the proton beam pulse onto four targets. The focusing of the secondary particles (mostly pions) produced by the protons impinging on the solid target inside the decay tunnel of the facility is realized by a hadronic collector system, which is based on four magnetic horns. Figure 2 (top) shows a 3D drawing of the four-horn system. The magnetic field inside each horn is produced by a 350 kA current pulse, generated by a power supply unit (PSU), of 100 µs duration, circulating inside the horn body and repeated at 14 Hz. These high current intensity and frequency, as well as the emission of secondary particles, produce a significant amount of power deposition in the horns. The target technology is based on a granular target concept that is able to afford 1.25 MW proton beam power with 2.5 GeV proton kinetic energy. The target geometry consists of a packed bed target of 78 cm long and 3 cm diameter, filled with 3 mm diameter titanium spheres. The target canister is drilled with apertures on each side to provide an efficient transverse cooling with helium flow working at 10 bar to extract 138 kW of deposited power. Figure 2 (bottom) shows a 3D drawing of the packed-bed target design.

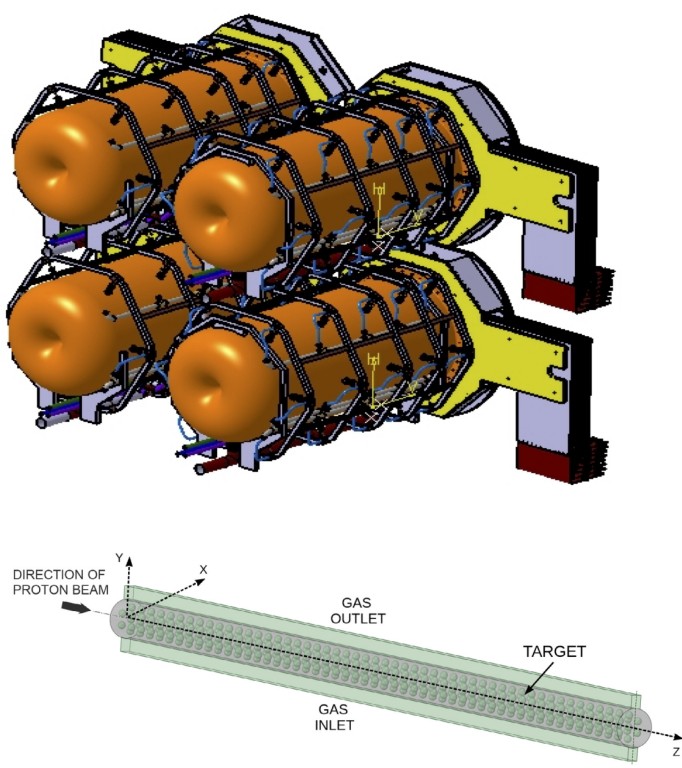

**Figure 2.** (**Top**) A 3D drawing of the four-horn target system. (**Bottom**) A 3D model of the target concept based on a packed bed of titanium spheres.

### 3. The ESS$\nu$SB+ Target Station Concept

For the new studies, a special transfer line from the ESS$\nu$SB ring-to-target transfer line will be added to extract the proton beam from the accumulator ring to a special target station of the ESS$\nu$SB+. This target station will contain one target-horn assembly, similar to that designed for the ESS$\nu$SB. It will feed the LEnuSTORM and the LEMNB with pions and muons. A pion and muon extraction system will be added to the ESS$\nu$SB+ target station. The horn shape will be re-optimized for this utilization. Figure 3 shows a conceptual drawing of the concept of the target station facility with the Pion Extraction and Initial Focusing System (PEIFS).

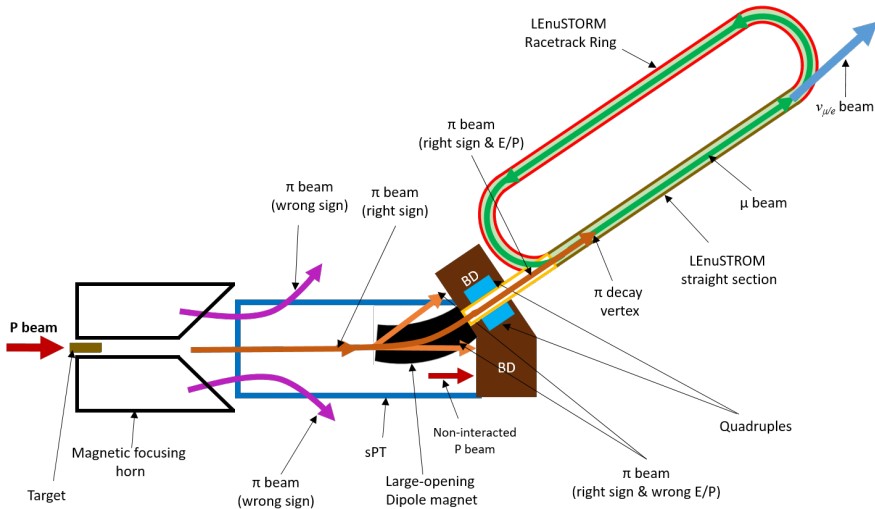

**Figure 3.** A conceptual drawing of the ESS$\nu$SB+ target station facility, with the concept of the Pion Extraction and Initial Focusing System.

The previous ESS$\nu$SB simulations have shown that the secondary beam, produced from the interaction between the proton beam and the target and focused by the horn, has an average momentum of 700 MeV. The beam profile follows a 2D Gaussian distribution with a beam width (FWHM) of more than ∼1 m at the beam dump level, i.e., 50 m from the horn exit. With such a beam momentum, the average decay length of the pions, $L_{\pi^{\pm}}$, is found to be ∼40 m. In order to select the proper length of the pion tunnel, where the pion extraction system will be placed in, dedicated preliminary simulations of the secondary beam radial distribution as a function of the distance, from 1 m to 10 m, from the horn exit was conducted in the preparation of the ESS$\nu$SB+ application. These simulations have shown that the secondary beam will follow a 2D Gaussian profile with $\sigma$∼0.38 m, at 10 m from the horn exit (Figure 4 (left)). Because extracting and focusing such a large beam is not a trivial process, a dedicated particle extracting and deviating system with proper "sufficiently" large aperture will need to be investigated. We have found a similar setup that uses a large-opening dipole magnet in the BGOOD experiment [11] at the ELSA facility in Bonn University, Germany, shown in Figure 4 (right). Such a magnetic system that depends on using conventional dipoles to bend the beam in both horizontal and vertical planes will be considered as a starting design in these new studies. Moreover, sets of magnetic focusing elements, e.g., quadrupoles, will ensure keeping the transverse envelops of the extracted pion beam minimal while propagating through the beam line. Furthermore, serious safety problems can be encountered due to the concomitant increase in the radiation levels, requiring substantially more shielding and protection. Therefore, a dedicated comprehensive radio-safety simulation devoted to detailed calculations of the different parts of the target station facility activation will be applied in order to precisely determine the shielding characteristics, e.g., thickness, required to comply with the radiological regulations of the site.

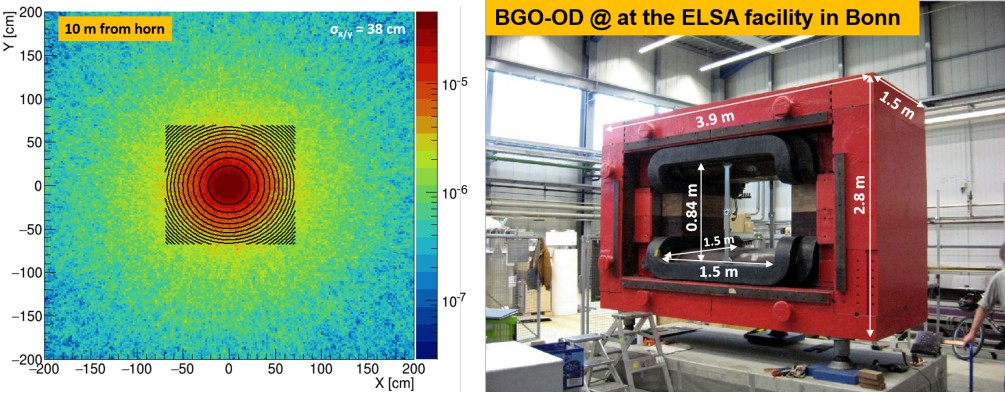

**Figure 4.** (**Left**) Secondary beam distribution, with 2D Gaussian fit (black circles), at 10 m from the horn exit. (**Right**) BGOOD dipole magnet at the ELSA facility in Bonn University [11].

## 4. Conclusions

After the success of the H2020 ESS$\nu$SB design study, proving the feasibility of the upgrade of the European Spallation Source to become, in addition to a neutron facility, also a very competitive neutrino facility, an extension project, the ESS$\nu$SB+, has been proposed— and accepted—to reinforce and develop complementary studies of the original physics goal of the project to study the CPV in the leptonic sector. A dedicated target station equipped with a pion extraction and focusing system will be designed within the framework of this new design study.

**Author Contributions:** Conceptualization, T.T. and E.B.; methodology, T.T. and E.B.; investigation, T.T. and E.B.; writing—original draft preparation, T.T.; writing—review and editing, T.T. and E.B. All authors have read and agreed to the published version of the manuscript.

**Funding:** This research was funded by the European Union's Horizon 2020 research and innovation program under grant agreement No. 777419, the agencies CNRS/IN2P3–France, and the Deutsche Forschungsgemeinschaft (DFG, German Research Foundation)—Projektnummer 423761110.

**Institutional Review Board Statement:** Not applicable.

**Informed Consent Statement:** Not applicable.

**Data Availability Statement:** Not applicable.

**Acknowledgments:** This project has been supported by the COST Action EuroNuNet "Combining forces for a novel European facility for neutrino-antineutrino symmetry-violation discovery" and by the European Union's Horizon 2020 research and innovation program under grant agreement No. 777419. We also acknowledge support of the funding agencies CNRS/IN2P3–France, the Deutsche Forschungsgemeinschaft (DFG, German Research Foundation)—Projektnummer 423761110, the Bulgarian National Science Fund Contract DCOST01/8, the Ministry of Science and Education of Republic of Croatia, grant No. KK.01.1.1.01.0001, and the Spanish Agencia Estatal de Investigacion through the grants IFT Centro de Excelencia Severo Ochoa No. CEX2020-001007-S and PID2019-108892RB, funded by MCIN/AEI/10.13039/501100011033. The authors would like to thank Tom Jude from the BGOOD experiment in Bonn University for discussions and supporting documents on the large-opening dipole magnet.

**Conflicts of Interest:** The authors declare no conflict of interest.

**Abbreviations**

The following abbreviations are used in this manuscript:

| | |
|---|---|
| CDR | Conceptual design report |
| CPV | Charge-parity violation |
| ESS$\nu$SB | European Spallation Source neutrino Super Beam |
| ESS$\nu$SB+ | European Spallation Source neutrino Super Beam plus |
| LEnuSTORM | Low-Energy neutrinos from Stored Muons |
| LEMNB | Low-Energy Monitored Neutrino Beam |
| LINAC | Linear accelerator |
| PEIFS | Pion Extraction and Initial Focusing System |
| R2S | Ring-to-switchyard |
| BSY | Beam switchyard |
| PSU | Power supply unit |
| BGOOD | BGO open dipole |

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
