# Peer review of "ESSνSB+ Target Station Concept†"

_psf, doi:10.3390/psf8010057_

Round 1

Reviewer 1 Report

The manuscript provides a good overview of the ESSnSB+ project and its future prospects. In general, the proceeding is clearly written and is of high interest for the neutrino community, also showing the great potential of this experiment and the different facilities involved. Therefore, I recommend its publication after considering some minor questions and suggestions that can be found below these lines and that could make the article more readable and self-contained.

Comments and suggestions:

- In page 1 CDR is mentioned but it is not defined in the Abbreviations' list at the end of the document.

- In page 1 \nu-N is used to describe neutrino-nucleus interactions, however nu-N is generally used to describe neutrino-nucleon interactions, nu-A would be more appropriate in this case, being A a generic nucleus.

- Different facilities are described in the manuscript. It could be interesting to mention in the text the place/country of the proposed facilities.

- At the beginning of page 2, it is stated that a short description of neutrino cross-section studies will also be given. If appropriate, it could be specified in which section or sections these new studies will be described.

- The definition of LINAC (page 2) could be added in the Abbreviations' list.

- Page 3. line 88. There is a typo: this --> these

Author Response

At first, I would like to thank the reviewer very much for his/her efforts in reviewing and perfecting the manuscript.

Here are the responses to the reviewer comments:

- In page 1 CDR is mentioned but it is not defined in the Abbreviations' list at the end of the document

Done

- In page 1 \nu-N is used to describe neutrino-nucleus interactions, however nu-N is generally used to describe neutrino-nucleon interactions, nu-A would be more appropriate in this case, being A a generic nucleus.

Done

- Different facilities are described in the manuscript. It could be interesting to mention in the text the place/country of the proposed facilities.

Indeed, there were some facilities, which their hosting countries are not mentioned in the text, I have completed those now. On the other hand, most of the facilities mentioned in the text, e.g. DUNE, T2HK and ESSnuSB, are long baseline experiments, where the neutrino source is located at some place/city while the detector is located at another city, few hundred kilometers away from the source. This makes it hard to mention a specific place/city of these experiments (but the countries are mentioned). As for the LEMNB and LEnuSTORM facilities, since they are a proposed additions to the ESSvSB experiment, which is, as mentioned in the first paragraph of the introduction, will be located in Sweden, I didn't specify their locations again in the text.

- At the beginning of page 2, it is stated that a short description of neutrino cross-section studies will also be given. If appropriate, it could be specified in which section or sections these new studies will be described.

I couldn't find this statement! Or maybe I misunderstand the comment! However, a reference to the existing nu-A cross-section studies at higher and intermediate energies is included in the text. 

- The definition of LINAC (page 2) could be added in the Abbreviations' list.

Done

- Page 3. line 88. There is a typo: this --> these

Done